# One out of four patients with pancreatic cancer experience psychological symptoms: A systematic review and meta-analysis

**Anna Sára Bognár[1], Dóra Andrea Sükös[1], Cristina Patoni[1,2], Eszter Ágnes Szalai[1,3], Brigitta Teutsch[1,4,5], Dániel Sándor Veres[1,6], Nóra Hosszúfalusi[1,7], Bálint Mihály Erőss[1,5,8], Katalin Márta[1,8,9☯], Péter Hegyi** [1,5,8,9☯]*

**1** Centre for Translational Medicine, Semmelweis University, Budapest, Hungary, **2** Carol Davila University of Medicine and Pharmacy, Bucharest, Romania, **3** Department of Restorative Dentistry and Endodontics, Semmelweis University, Budapest, Hungary, **4** Department of Radiology, Medical Imaging Centre, Semmelweis University, Budapest, Hungary, **5** Institute for Translational Medicine, Szentágothai Research Centre, Medical School, University of Pécs, Pécs, Hungary, **6** Department of Biophysics and Radiation Biology, Semmelweis University, Budapest, Hungary, **7** Department of Internal Medicine and Hematology, Semmelweis University, Budapest, Hungary, **8** Institute of Pancreatic Diseases, Semmelweis University, Budapest, Hungary, **9** Translational Pancreatology Research Group, Interdisciplinary Centre of Excellence for Research Development and Innovation, University of Szeged, Szeged, Hungary

☯ These authors contributed equally to this work.
* hegyi2009@gmail.com

## Abstract

The death rate from pancreatic cancer is one of the highest of all cancers, with increasing incidence every year over the past decades. As the prognosis remains extremely poor, this trend brings not only a growing clinical challenge, but also a significant psychological toll for patients. In comparison to other types of cancer, our understanding of the psychological burden of such a deadly disease is still limited. Recognizing and addressing the psychological aspects of pancreatic cancer are crucial for comprehensive patient care. We, therefore, aimed to quantify the psychological burden of patients with pancreatic cancer. We conducted a systematic search (PROSPERO: CRD42022288805) of MEDLINE, Cochrane, and Embase databases to identify articles reporting on the prevalence of psychological symptoms among patients with pancreatic cancer. The symptoms examined were depression, anxiety, distress, fatigue, sleep disturbances. Study quality for the included studies was assessed using the Joanna Briggs Institute tool. Relevant data were extracted and exported for quantitative synthesis. The meta-analysis of the proportion of symptoms with a 95% confidence interval (CI) was conducted using frequentist random-effects models. Subset analyses were performed based on the severity of depression. A total of 28 articles were included in the meta-analysis and 31 in the systematic review. Our results showed that the proportion of depression (19 studies; 78,930 patients) was 0.27 (95% CI: 0.18–0.39). The proportion for anxiety (18 studies; 26,538 patients) was 0.29 (95% CI: 0.16–0.46; for distress (6 studies; 2746 patients) 0.43 (95% CI: 0.25–0.63); for fatigue (7 studies; 11 338 patients) 0.40 (95% CI: 0.21–0.63); and

**Data availability statement:** All data extracted for the meta-analysis are included in the paper tables, figures and Supporting Information files.

**Funding:** The research was supported by the Hungarian Ministry of Innovation and Technology, National Research, Development and Innovation Fund (TKP2021-EGA-23 to PH). New National Excellence Program of the Ministry for Innovation and Technology from the source of the National Research, Development and Innovation Fund (to BT - ÚNKP-23-3-II-PTE-1996). Sponsors had no role in the design, data collection, analysis, interpretation, and preparation of the manuscript.

**Competing interests:** The authors have declared that no competing interests exist.

for sleep disturbances (3 studies; 10,645) 0.31 (95% CI: 0.06–0.77). These findings highlight the urgent need to integrate psychological care into pancreatic cancer treatment. Future research should explore sources of heterogeneity to better identify which approaches provide the most psychological benefit and tailor interventions to patient needs accordingly.

## Introduction

Pancreatic cancer (PC) is currently the third deadliest cancer type in the United States, it is projected that approximately 67,440 new cases will be diagnosed in 2025, leading to the loss of 51,980 lives [1]. The prevalence of PC has been steadily increasing, making it a substantial public health concern, [1] and by 2040, the estimated number of incidents is expected to almost double [2]. The same trend can be seen across Europe, where PC is the only cancer whose incidence and mortality rates are rising in both sexes [3]. This will pose significant clinical challenges and have profound psychological implications for patients, their families, and healthcare professionals due to the disease's poor prognosis, rapid progression, and high symptom burden.

Research has demonstrated that elevated levels of depression and anxiety can negatively impact quality of life, predict poorer adherence to treatment regimens, and are independently associated with decreased overall survival rates in this patient population [4,5]. It has been also shown earlier that psychological interventions have beneficial effects on patients with cancer and can improve their quality of life at the onset of their disease [6]. Despite these findings, the International Psycho-Oncology Society (IPOS) [7] has found significant differences in the delivery of psycho-oncology services. In Europe, 40–60% of patients with cancer and their families face psychological distress that could be treated with adequate interventions. However, only a minority of individuals in this group receive the psychological support and care they need [8]. Indeed, only 37% of European countries have an allocate budget for psycho-oncological support, and in most Eastern European countries, mental health remains a low priority. On the contrary, addressing psychological issues are still stigmatized, and their impact is overlooked [9].

As the prevalence of PC increases, mental health problems need to be addressed even more, and we need to map patients' psychological needs to personalize and provide the best possible care. Although individual studies may have previously reported on the psychological symptoms of patients with PC, there is still a need for a comprehensive overview that can initiate future research, clinical practice, and interventions in this specific oncological context. This meta-analysis aims to consolidate available evidence and quantify the burden of psychological symptoms in patients with PC, as assessed by screening measures rather than clinically diagnosed psychiatric disorders.

## Methods

For this systematic review and meta-analysis, we followed the recommendations of the Cochrane Collaboration and Preferred Reporting Items for Systematic Reviews

and Meta-Analyses (PRISMA) reporting guidelines [10]. The PRISMA checklist can be found in the Supporting Information, Table 1 (S1 Table). The review protocol was registered in the International Prospective Register of Systematic Reviews (PROSPERO; CRD42022288805), and no deviations from the original protocol occurred.

## Eligibility criteria

Our clinical question was what is the prevalence of psychological symptoms among patients with pancreatic cancer? We formulated our clinical question and defined eligibility criteria for our review using the CoCoPop (Condition, Context, Population) framework. We included full articles reporting the prevalence of any psychological symptoms among adult patients with diagnosed pancreatic ductal adenocarcinoma, regardless of stage. The search was restricted to peer-reviewed publications; grey literature was not included. For the purposes of this review, outcomes were conceptualized as psychological symptom burden rather than prevalence of clinically diagnosed psychiatric disorders. Given that the term 'psychological symptoms' is a vague term widely used in the literature, any definition was accepted. The definitions of depression, anxiety, distress, fatigue, and sleep disturbances varied across studies and were often vague. We accepted them if they were based on standardized diagnostic criteria, or if they are commonly used in clinical and research settings with their specified cut-off scores in the articles. For the subgroup analysis on severity, we used the categorization of the original articles (mild, moderate, severe). The exclusion criterion was any premorbid psychiatric disease. Case reports and case series were not considered eligible; however, any other type of article was eligible.

## Search strategy

The search was conducted in MEDLINE, EMBASE, and Cochrane Library on May 22, 2022, and updated on March 3, 2025. There was no date or language restriction, for non-English articles, when none of the co-authors were fluent in the language, we used online translation tools to assess eligibility and extract relevant data. The search key can be found in in the Supporting Information in S1 Documentum. We reviewed the reference lists of identified relevant studies to see if any additional relevant studies might be eligible for our studies.

## Selection process

Duplicates were removed using reference manager software (EndNote X9, Clarivate Analytics, Philadelphia, PA, USA). Titles, abstracts, and, later, full texts were selected by two independent authors in pairs (DAS and CP) based on pre-specified criteria. Any possible disagreements were resolved by a third party (SAB). Cohen's kappa coefficients were calculated to measure inter-rater reliability.

## Data collection process

Two authors (DAS and SAB) independently collected data from the eligible articles manually into a preformed, standardized data table (Microsoft Excel, Microsoft Office 365, Redmond, WA, USA). Disagreements were resolved by a third person (CP).

## Data items

The following data were extracted: publication characteristics: first author, year of publication, country of origin, number of centers, demographic characteristics of the study population, measurement tool, time of measurement (referred to as time groups), occurrence of psychological symptoms as reported in each article. Prevalence was estimated as the proportion of PC patients affected by a psychological symptom at any given time.

## Study risk of bias assessment

Two independent reviewers (DAS and SAB) assessed the methodological quality of the studies using The Joanna Briggs Institute Prevalence Critical Appraisal Tool [11]. A third reviewer (CP) resolved potential disagreements about the risk of

bias scoring. The tool includes nine questions on study design, duration of the study, retention of participants, and study results. A 'yes' option is provided to indicate appropriate methodological quality. Alternatively, a 'no' option can be selected if poor quality is indicated or 'unclear' for ambiguous responses, improving the assessment of study quality. Each item is assessed using a score (yes = 1), (no = 0), and (unclear = 0). The total score from each study was also presented as a percentage. Different levels of risk of bias are categorized as follows: high risk of bias = 20–50% items scored yes; moderate risk of bias = 50–80% items scored yes; and low risk of bias = 80–100% items scored yes, based on the JBI checklist.

### Effect measures and synthesis methods

As we assumed considerable between-study heterogeneity in all cases, a random-effects model was used to pool effect sizes in a frequentist method. The proportion (prevalence) of symptoms with a 95% confidence interval (CI) was used for effect size measure. The total number of patients and those with the event of interest were extracted from each study to calculate the proportion. We summarized the findings from the meta-analysis in forest plots. We also reported the prediction intervals (i.e., the expected range of effects of future studies) of results, where applicable: the number of studies was large enough (>5), and the effects were not too heterogeneous. Between-study heterogeneity was described by the between-study variance ($\tau^2$) and Higgins and Thompson's $I^2$ statistics [12]. Results were reported in the format of effect size or statistics and [95% CI]. Small study publication bias was assessed by visual inspection of funnel plots and calculating modified Egger's test p-values. However, we kept in mind that the test had limited diagnostic assessment below ~10 studies. Potential outlier publications were detected using different influence measures and plots following the recommendation by Harrer et al.[13]. All statistical analyses were calculated by R software [14] (v4.3.3) using the meta [15] (v7.0.0) package for basic meta-analysis calculations and plots and the dmetar [16] (v0.1.0) package for additional influential analysis calculations and plots. For more details on data synthesis methods and influential analysis, see 'Data synthesis – detailed methods' in the Supporting Information, in S2 Documentum.

### Ethical approval

No ethical approval was required for this systematic review with meta-analysis, as all data were already published in peer-reviewed journals. No patients were involved in the design, conduct or interpretation of our study.

Datasets used in this study can be found in the full-text articles included in the systematic review and meta-analysis.

## Results

### Search and selection

Altogether, 23,282 studies were identified using our search key. After duplicate removal and exclusions based on titles and abstracts, 313 papers were screened for eligibility. After assessing the full text, we excluded 194 articles, mainly due to ineligible study design (n = 12), no data on pancreatic cancer outcomes (n = 32), or no prevalence data (n = 44), resulting in 28 papers being used in the meta-analysis and 31 in the systematic review section. The whole process is described in the PRISMA flowchart Fig 1. The list of excluded studies is provided in S4 Documentum.

### Basic characteristics of studies included in the meta-analysis

The baseline characteristics of the analyses included are detailed in S2 Table. In summary, 19 studies [4,17–31] reported on depression, 18 studies [4,17–19,21–23,26,27,30–37] on anxiety, 6 studies [21,29,38–41] on distress, 7 studies [4,21,27,29,32,37,42] on fatigue, and three studies on sleep disturbances [4,29,31]. The number of participants ranged from 11 to 62,450. The baseline characteristics of the three additional studies [43–45] in the systematic review are detailed in S3 Table. Their reported results showed similar prevalence to the ones in the meta-analysis.

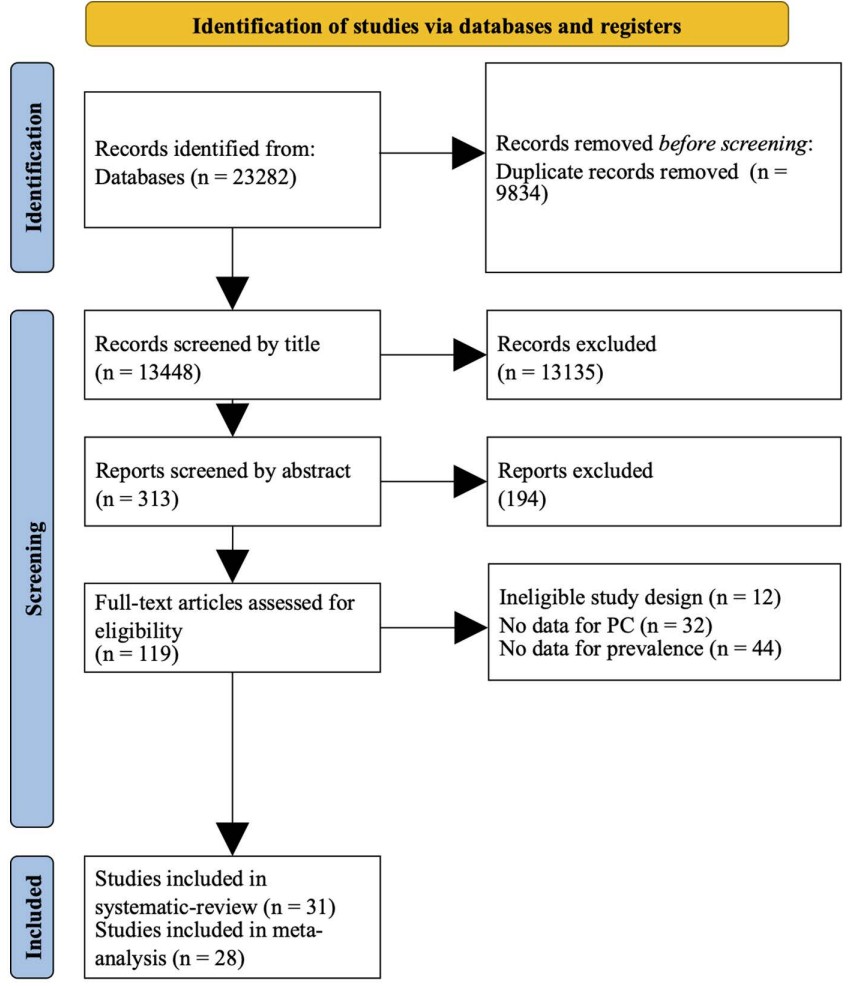

**Fig 1. PRISMA Flowchart.**

## Prevalence of psychological symptoms among patients with PC

**Depression.** For depression, 78,930 patients from 19 studies were involved in our analysis. Our results showed that the proportion was 0.27 (95% CI: 0.18–0.39); $I^2 = 0.982$ (95% CI: 0.978–0.985). In the studies where data based on severity was available the subset analysis based on severity showed that the proportion of mild depression was 0.26 (95% CI: 0.16–0.38), based on five studies; the proportion of moderate depression was 0.28 (95% CI: 0.11–0.55), based on five studies; and the proportion of severe depression was 0.16 (95% CI: 0.11–0.22), based on six studies (Fig 2).

### Anxiety

Eighteen studies involving 26,538 patients were included in our analysis of anxiety. Our results showed that the proportion of patients with anxiety was 0.29 (95% CI: 0.16–0.46); $I^2 = 0.994$ (95% CI: 0.994–0.995). (Fig 3).

### Distress

Data from 2746 patients from six studies were analyzed for distress. Our results showed that the proportion of distress was 0.43 (95% CI: 0.25–0.63); $I^2 = 0.966$ (95% CI: 0.945–0.978). (Fig 4).

| Study | Follow-up Time | Patients with Depression | Total Number of Patients | Proportion of Depression | Proportion | 95% CI | ROB |
|---|---|---|---|---|---|---|---|
| Akizuki, et al., 2016 | short | 3 | 110 | | 0.03 | [0.01; 0.08] | low |
| Brintzenhofe, et al., 2009 | | 18 | 185 | | 0.10 | [0.06; 0.15] | low |
| Yeo, et al., 2023 | | 49 | 403 | | 0.12 | [0.09; 0.16] | low |
| Seoud, et al., 2020 | long | 8130 | 62450 | | 0.13 | [0.13; 0.13] | low |
| Harris, et al., 2021 | long | 1581 | 10378 | | 0.15 | [0.15; 0.16] | moderate |
| Godby, et al., 2020 | short | 14 | 88 | | 0.16 | [0.10; 0.25] | low |
| Salm, et al., 2021 | long | 16 | 80 | | 0.20 | [0.13; 0.30] | low |
| Subramaniam, et al., 2024 | short | 1034 | 4029 | | 0.26 | [0.24; 0.27] | low |
| Vehling, et al., 2022 | long | 14 | 50 | | 0.28 | [0.17; 0.42] | low |
| Clark, et al., 2010 | short | 87 | 304 | | 0.29 | [0.24; 0.34] | low |
| Batra, et al., 2021 | long | 27 | 94 | | 0.29 | [0.21; 0.39] | moderate |
| Janda, et al., 2017 | long | 42 | 136 | | 0.31 | [0.24; 0.39] | low |
| Hartung, et al., 2017 | short | 28 | 82 | | 0.34 | [0.25; 0.45] | low |
| Pezzili, et al., 2017 | short | 8 | 22 | | 0.36 | [0.20; 0.57] | moderate |
| Del Piccolo, et al., 2021 | short | 180 | 400 | | 0.45 | [0.40; 0.50] | moderate |
| Fras, et al., 1967 | short | 23 | 50 | | 0.46 | [0.33; 0.60] | high |
| Boyd, et al., 2012 | long | 13 | 22 | | 0.59 | [0.39; 0.77] | moderate |
| Kim, et al., 2023 | short | 25 | 36 | | 0.69 | [0.53; 0.82] | low |
| Carruba, et al., 2022 | | 10 | 11 | | 0.91 | [0.60; 1.00] | low |
| **Pooled (random) effect** | | | | | **0.27** | **[0.18; 0.39]** | |
| **Prediction interval** | | | | | | **[0.04; 0.78]** | |

$I^2 = 98.2\%$ [97.8%; 98.5%]  $\tau = 1.03$

| Study | Follow-up Time | Patients with Depression | Total Number of Patients | Proportion of Mild Depression | Proportion | 95% CI | ROB |
|---|---|---|---|---|---|---|---|
| Janda, et al., 2017 | long | 22 | 136 | | 0.16 | [0.11; 0.23] | low |
| Pezzili, et al., 2017 | short | 5 | 22 | | 0.23 | [0.10; 0.44] | moderate |
| Hartung, et al., 2017 | short | 25 | 82 | | 0.30 | [0.22; 0.41] | low |
| Boyd, et al., 2012 | long | 7 | 22 | | 0.32 | [0.16; 0.53] | moderate |
| Kim, et al., 2023 | short | 13 | 36 | | 0.36 | [0.22; 0.52] | low |
| **Pooled (random) effect** | | | | | **0.26** | **[0.16; 0.38]** | |

$I^2 = 58.9\%$ [0.0%; 84.7%]  $\tau = 0.32$

| Study | Follow-up Time | Patients with Depression | Total Number of Patients | Proportion of Moderate Depression | Proportion | 95% CI | ROB |
|---|---|---|---|---|---|---|---|
| Pezzili, et al., 2017 | short | 2 | 22 | | 0.09 | [0.01; 0.29] | moderate |
| Kim, et al., 2023 | short | 6 | 36 | | 0.17 | [0.07; 0.32] | low |
| Boyd, et al., 2012 | long | 5 | 22 | | 0.23 | [0.10; 0.44] | moderate |
| Hartung, et al., 2017 | short | 40 | 82 | | 0.49 | [0.38; 0.59] | low |
| Carruba, et al., 2022 | | 6 | 11 | | 0.55 | [0.28; 0.79] | low |
| **Pooled (random) effect** | | | | | **0.28** | **[0.11; 0.55]** | |

$I^2 = 79.5\%$ [51.5%; 91.3%]  $\tau = 0.79$

| Study | Follow-up Time | Patients with Depression | Total Number of Patients | Proportion of Severe Depression | Proportion | 95% CI | ROB |
|---|---|---|---|---|---|---|---|
| Boyd, et al., 2012 | long | 1 | 22 | | 0.05 | [0.00; 0.24] | moderate |
| Pezzili, et al., 2017 | short | 1 | 22 | | 0.05 | [0.00; 0.24] | moderate |
| Janda, et al., 2017 | long | 20 | 136 | | 0.15 | [0.10; 0.22] | low |
| Kim, et al., 2023 | short | 6 | 36 | | 0.17 | [0.07; 0.32] | low |
| Hartung, et al., 2017 | short | 17 | 82 | | 0.21 | [0.13; 0.31] | low |
| Carruba, et al., 2022 | | 4 | 11 | | 0.36 | [0.15; 0.65] | low |
| **Pooled (random) effect** | | | | | **0.16** | **[0.11; 0.22]** | |

$I^2 = 37.9\%$ [0.0%; 75.3%]  $\tau = 0$

**Fig 2. Prevalence of depression.** The forest plot shows the prevalence of depression total following a pancreatic cancer diagnosis using subgroup analysis based on severity (mild, moderate, severe). 'Short'-within six months of cancer diagnosis; 'long'-beyond six months from cancer diagnosis. ROB- Risk of bias, CI-Confidence interval.

| Study | Follow-up Time | Patients with Anxiety | Total Number of Patients | Proportion of Anxiety | Proportion | 95% CI | ROB |
|-------|------|------|------|------|------|------|------|
| Akizuki, et al., 2016 | short | 2 | 110 | | 0.02 | [0.00; 0.07] | low |
| Mehnert, et al. , 2014 | short | 3 | 52 | | 0.06 | [0.01; 0.16] | low |
| Salm, et al. ,2021 | long | 6 | 80 | | 0.07 | [0.03; 0.16] | low |
| Vehling, et al. ,2022 | long | 4 | 50 | | 0.08 | [0.03; 0.19] | low |
| Brinzenthofe , et al., 2009 | | 17 | 185 | | 0.09 | [0.06; 0.14] | low |
| Harris, et al., 2021 | long | 1167 | 10378 | | 0.11 | [0.11; 0.12] | moderate |
| Janda, et al. , 2017 | long | 20 | 136 | | 0.15 | [0.10; 0.22] | low |
| Subramaniam, et al. ,2024 | short | 1068 | 4029 | | 0.27 | [0.25; 0.28] | low |
| Clark, et al., 2010 | short | 89 | 304 | | 0.29 | [0.24; 0.35] | low |
| Batra, et al. , 2021 | long | 29 | 94 | | 0.31 | [0.22; 0.41] | moderate |
| Del Piccolo, et al. , 2021 | short | 180 | 400 | | 0.45 | [0.40; 0.50] | moderate |
| Fras, et al. , 1967 | short | 16 | 35 | | 0.46 | [0.30; 0.62] | high |
| Seoud, et al. , 2020 | long | 4740 | 10220 | | 0.46 | [0.45; 0.47] | low |
| Lelond, et al. , 2021 | short | 62 | 123 | | 0.50 | [0.42; 0.59] | moderate |
| Zhang, et al. ,2022 | | 53 | 100 | | 0.53 | [0.43; 0.62] | low |
| Cui, et al. ,2023 | | 132 | 209 | | 0.63 | [0.56; 0.69] | low |
| Carruba, et al. ,2022 | | 9 | 11 | | 0.82 | [0.51; 0.96] | low |
| Boyd, et al. , 2012 | long | 21 | 22 | | 0.95 | [0.76; 1.00] | moderate |
| **Pooled (random) effect** | | | | | 0.29 | [0.16; 0.46] | |
| **Prediction interval** | | | | | | [0.01; 0.91] | |

$I^2 = 99.4\%$ [99.4%; 99.5%]   $\tau = 1.51$

**Fig 3. Prevalence of anxiety.** The forest plot shows the prevalence of anxiety following a diagnosis of pancreatic cancer. 'Short'-within six months of cancer diagnosis; 'long'-beyond six months from cancer diagnosis. ROB- Risk of bias, CI-Confidence interval.

## Fatigue

Seven studies reporting fatigue with 11,338 patients were involved in our analysis. Our results showed that the prevalence was 0.40 (95% CI: 0.21–0.63); $I^2 = 0.973$ (95% CI: 0.959–0.982). (Fig 5).

## Sleep disturbances

Three studies reporting sleep disturbances with 10,645 patients were involved in our analysis. Our results showed that the prevalence was 0.31 (95% CI: 0.06–0.77); $I^2 = 0.981$ (95% CI: 0.965–0.990). (Fig 6).

## Assessment of methodological quality

The quality of the included studies was assessed using the Joanna Briggs Institute Critical Appraisal Tool. Studies were grouped by risk of bias (high, medium, low), and these groups are also reflected in the forest plots. Among the 28 included studies, 21 were rated as low risk of bias, six as moderate risk, and one as high risk according to the JBI tool. Detailed scores and assessment outcomes are provided in Supporting Information, S4 Table.

## Publication bias, heterogeneity, and sensitivity analysis

We provided the funnel plot analysis with p-values for depression and anxiety only (S1 Fig,2), as we were not given a funnel plot assessment when the study number was below ten. Egger's test p-value is 0.0413 for depression indicating potential publication bias. Egger's test p-value is 0.7877 for anxiety indicating no potential publication bias. High heterogeneity was observed across studies, with $I^2$ values exceeding 90% for all measured outcomes, indicating substantial variability between studies. The Leave-one-out analysis revealed that the potential influential studies have no relevant impact

| Study | Follow-up Time | Patients with Distress | Total Number of Patients | Proportion of Distress | Proportion | 95% CI | ROB |
|---|---|---|---|---|---|---|---|
| Clark, et al., 2010 | short | 80 | 304 | | 0.26 | [0.22; 0.32] | low |
| Dai, et al., 2019 | short | 572 | 2043 | | 0.28 | [0.26; 0.30] | low |
| Hohmann, et al. ,2022 | | 4 | 11 | | 0.36 | [0.15; 0.65] | low |
| Carlson, et al., 2004 | short | 41 | 112 | | 0.37 | [0.28; 0.46] | low |
| Carlson, et al., 2019 | short | 89 | 148 | | 0.60 | [0.52; 0.68] | low |
| Yeo, et al. ,2023 | | 93 | 128 | | 0.73 | [0.64; 0.80] | low |
| **Pooled (random) effect** | | | | | **0.43** | **[0.25; 0.63]** | |
| **Prediction interval** | | | | | | **[0.09; 0.86]** | |

$I^2 = 96.6\%$ [94.5%; 97.8%]   $\tau = 0.74$

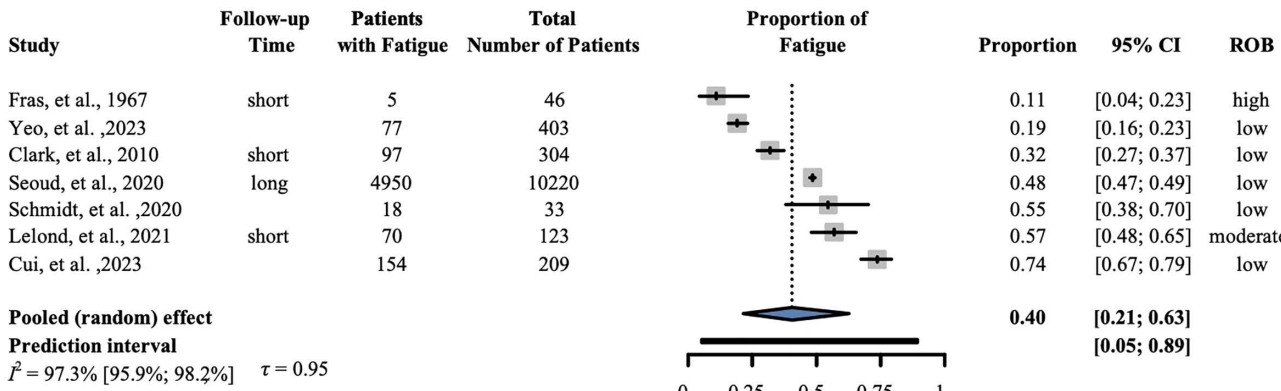

**Fig 4. Prevalence of distress.** The forest plot shows the prevalence of distress following a diagnosis of pancreatic cancer. 'Short'-within six months of cancer diagnosis. ROB- Risk of bias, CI-Confidence interval.

| Study | Follow-up Time | Patients with Fatigue | Total Number of Patients | Proportion of Fatigue | Proportion | 95% CI | ROB |
|---|---|---|---|---|---|---|---|
| Fras, et al., 1967 | short | 5 | 46 | | 0.11 | [0.04; 0.23] | high |
| Yeo, et al. ,2023 | | 77 | 403 | | 0.19 | [0.16; 0.23] | low |
| Clark, et al., 2010 | short | 97 | 304 | | 0.32 | [0.27; 0.37] | low |
| Seoud, et al., 2020 | long | 4950 | 10220 | | 0.48 | [0.47; 0.49] | low |
| Schmidt, et al. ,2020 | | 18 | 33 | | 0.55 | [0.38; 0.70] | low |
| Lelond, et al., 2021 | short | 70 | 123 | | 0.57 | [0.48; 0.65] | moderate |
| Cui, et al. ,2023 | | 154 | 209 | | 0.74 | [0.67; 0.79] | low |
| **Pooled (random) effect** | | | | | **0.40** | **[0.21; 0.63]** | |
| **Prediction interval** | | | | | | **[0.05; 0.89]** | |

$I^2 = 97.3\%$ [95.9%; 98.2%]   $\tau = 0.95$

**Fig 5. Prevalence of fatigue.** The forest plot shows the prevalence of fatigue following a diagnosis of pancreatic cancer. 'Short'-within six months of cancer diagnosis; 'long'-beyond six months from cancer diagnosis. ROB- Risk of bias, CI-Confidence interval.

| Study | Follow-up Time | Patients with Sleep Disturbances | Total Number of Patients | Proportion of Sleep Disturbances | Proportion | 95% CI | ROB |
|---|---|---|---|---|---|---|---|
| Yeo, et al. ,2023 | | 55 | 403 | | 0.14 | [0.11; 0.17] | low |
| Seoud, et al., 2020 | long | 4230 | 10220 | | 0.41 | [0.40; 0.42] | low |
| Boyd, et al., 2012 | long | 11 | 22 | | 0.50 | [0.31; 0.69] | moderate |
| **Pooled (random) effect** | | | | | **0.31** | **[0.06; 0.77]** | |

$I^2 = 98.1\%$ [96.5%; 99.0%]   $\tau = 0.77$

**Fig 6. Prevalence of sleep disturbances.** The forest plot shows the prevalence of sleep disturbances following a diagnosis of pancreatic cancer. 'Short'-within six months of cancer diagnosis; 'long'-beyond six months from cancer diagnosis. ROB- Risk of bias, CI-Confidence interval.

on the pooled result (S3 Fig,8). This analysis also confirms that the high observed heterogeneity is multifactorial and not caused by a single influential outlier study

## Discussion

The purpose of this systematic review and meta-analysis was to investigate the prevalence of psychological symptoms among patients with PC. Our findings showed that the examined outcomes developed in more than a quarter of the studied population (depression 27%, anxiety 29%, distress 43%, fatigue 40%, and sleep disturbances 31%). Although we were not able to compare the effect of time from diagnosis (categorized as 'short'—within six months of cancer diagnosis, and 'long'—beyond six months from cancer diagnosis), the forest plots suggest that the timing of diagnosis does not impact the development of these psychological symptoms in a relevant amount, allowing plenty of time for interventions.

Mood disturbances, especially depression and anxiety, are often presenting complaints in patients with PC, with an overall prevalence ranging from 30% to 50% [46]. This study indicates a significant occurrence of depression among patients with PC. Earlier studies have found that 30% of patients with PC report symptoms of clinical depression [21,47]. Our subset analysis of depression based on severity reveals that prevalence is variable. A potential reason for the lower prevalence in the severe group is the rapid deterioration of the patient's condition. These patients often require heavy medication, including sedation, to reduce symptom burden, particularly when admitted to the intensive care unit. Similarly, our results indicate that a substantial proportion of patients experience anxiety alongside their cancer diagnosis. Even in early-stage disease, anxiety is common before surgery and preoperative counseling [46]. In a systematic review comparing anxiety and depressive symptoms in adults with other cancers, patients with PC showed greater anxiety and depressive symptoms [48].

Sleep disturbance is a common symptom in patients with cancer [49]. Sleep disturbance and fatigue data specific to the population with PC are rare, but most studies report high fatigue levels at presentation and throughout treatment [50]. Our findings show that fatigue affects a significant portion of the population. Treatments for PC often lead to loss of skeletal muscle mass, which can increase fatigue and psychological distress [51]. The proportion of distress was also high in our findings and is similar to earlier reports in which psychological distress affected approximately 40% of patients diagnosed with PC [47,52].

Previous studies have suggested that symptoms of fatigue, anxiety, and depression are strongly interconnected [53,54]. Compared with healthy adults or population norms, adults with pancreatic cancer experienced a lower quality of life (QoL) in all domains. Additionally, when compared with patients with other types of cancer, those with PC reported poorer psychological QoL, with a significant strain on psychological well-being [48].

Understanding the complexity of underlying factors is crucial when interpreting prevalence. The real-life presence of psychological symptoms among patients reported in the meta-analysis may be even higher for many reasons. Psychological symptoms, such as depression, anxiety, and distress, might be underreported due to stigma or reluctance to address mental health issues [55]. Cancer patients, including those with pancreatic cancer, may be hesitant to acknowledge or seek help for emotional challenges, which may lead to an underestimation of the prevalence of the symptoms.

The high heterogeneity observed in the meta-analysis, with I² values consistently above 90% (e.g., 98.2% for depression), indicates substantial variation in the reported prevalence estimates. We acknowledge that the pooled results should therefore be interpreted with caution. The potential sources of this high heterogeneity are multifactorial: reflecting differences in 1., cultural and societal contexts, 2., study populations, and variations in 3., measurement tools and diagnostic criteria, all of which can influence reported prevalence estimates.

A more homogeneous population could provide a more accurate value for prevalence, as cultural and societal factors influence how individuals express and perceive psychological symptoms, as seen in the studies included with high or low prevalence, e.g., in the study by Akizuki et al [17]. Variations in cultural norms, attitudes toward mental health, and the availability of support services can contribute to differences in reported proportions across studies. Age is also an

important factor for patients with PC. The risk for distress in patients with cancer was almost double that of the general population [56]. However, studies found that older age groups (61 years and above) showed almost no differences in psychological distress between patients with cancer and the general population, with differences being higher in younger patients [56,57]. Even though PC is typically the disease of the elderly, the incidence is growing among the younger population [58–60].

The measurement tools included in the studies are widely used in clinical and research settings, however, they are arguably the most significant driver of heterogeneity. Information on the questionnaire per study can be found in S2 Table. Although they all offer standardized measures for assessing various psychological constructs such as depression, anxiety, distress, and fatigue, they also raise questions of comparability and might affect the results [61]. Variations in administration protocols, scoring methods, cultural adaptations, and severity across different studies can further complicate the comparability and interpretability of findings from these instruments [62,63]. Differences in administration protocols, such as self-report versus clinician-administered assessments, can introduce variability in responses due to factors like literacy, fatigue, or willingness to disclose sensitive information. Self-report questionnaires, such as the Hospital Anxiety and Depression Scale (HADS), the Beck Depression Inventory (BDI), and the Patient Health Questionnaire-9 (PHQ-9), often result in higher prevalence estimates because patients may feel more comfortable reporting symptoms [64]. In contrast, clinician-administered assessments, such as the Structured Clinical Interview for DSM Disorders (SCID) tend to report lower prevalence but provide more accurate diagnostic data [65]. Scoring methods and cut-off thresholds directly influence prevalence estimates; for instance, the Distress Thermometer (DT) uses lower thresholds for high sensitivity, resulting in higher prevalence for general distress [66], whereas instruments like the HADS provide lower, more specific estimates for depression by excluding confounding somatic items [67]. These factors, together contribute to the high heterogeneity observed across analyses (I² consistently >90%), which should be considered when interpreting pooled prevalence estimates. Therefore, the pooled prevalence should be interpreted as an average estimate across heterogeneous operational definitions rather than a precise prevalence for any single measurement approach.

Future research should aim to standardize the definition of these constructs and narrow the use of these measurement tools to achieve more subjective and comparative results in psychological research.

Previous studies [68–70] have highlighted the critical role of psychological factors on cancer recurrence and mortality in different types of cancer. However, our understanding of the psychological burden associated with pancreatic cancer remains limited compared to other types of cancer, perhaps because less attention has been paid to it. A recent study [71] has revealed that many common and deadly cancers, including pancreatic cancer, receive the least research funding relative to their prevalence and mortality. Conversely, breast cancer and pediatric cancers are well-funded despite their lower mortality. These findings align with previous research, including an analysis of European cancer research investment from 2016 to 2020, highlighting the disproportionate research [72].

A first step toward development in this area is to assess the presence and extent of psychological burdens in PCs. This knowledge gap is particularly critical given the potential negative implications of psychological factors on disease course [4], poorer survival, and higher cancer-specific mortality [5].

The relatively high occurrence of psychological symptoms in this study underscores the importance of integrating psychological care into the comprehensive treatment of pancreatic cancer. Treating patients with cancer should be a multidisciplinary approach, where the aim is not only to treat physical symptoms but psychological ones as well, as they can influence the course and outcome of the disease. Disparities in access in many countries and smaller towns remain challenging [9] – leaving patients, their families, and healthcare professionals struggling with the emotional complexities of these serious cases without support. Bridging these gaps in access to psychological care is not only imperative for the well-being of individual patients, but also a fundamental step toward improving the holistic quality of care for those facing the challenges of pancreatic cancer.

By providing a detailed synthesis of existing data, our findings not only fill a critical gap in the literature and could serve as a base for further research but also advocate for increased funding and resources dedicated to understanding and addressing the burdens faced by patients with PC.

## Strengths and limitations

The strengths of our analysis, to the best of our knowledge, are that this is the first and most comprehensive meta-analysis on the topic, employs rigorous methods, and incorporates a high number of included patients and different psychological constructs in one article.

As for the limitations, besides the low number of studies on each psychological symptom, differences in the assessment tools and criteria used across studies can impact the prevalence. The articles' definitions of symptoms were variable and vague, measured with different tools, and severity thresholds varied. These factors, together with cultural or language adaptations, differences in patient populations contribute to the high heterogeneity observed across analyses (I² consistently >90%), which should be considered when interpreting pooled prevalence estimates. Additionally, funnel plots were only presented for depression and anxiety; other outcomes could not be assessed due to smaller sample sizes, which may limit the evaluation of potential publication bias for these outcomes.

## Implications for practice, research and policymakers

Addressing psychological symptoms, including depression, anxiety, distress, fatigue, and sleep disturbances is necessary for holistic patient care, and the translation of new scientific findings into clinical practice is crucial [73,74]. The findings of this study, which reveal a relatively high prevalence of psychological symptoms among patients with pancreatic cancer, underscore the critical need to integrate psychological support and interventions into the comprehensive care of these patients. This is particularly significant when considered alongside our previous research, which has demonstrated that psychological interventions can substantially enhance the QoL for patients, especially with early-stage cancer [6]. Routine psychological screening should be implemented for all patients with pancreatic cancer using brief, easy-to-administer tools such as the Distress Thermometer (DT) for general distress [75,76] and the Patient Health Questionnaire-9 (PHQ-9) or the Hospital Anxiety and Depression Scale (HADS),for depression [77]. While these questionnaires are practical for clinical use, their diagnostic accuracy is limited and cannot replace a complete clinical assessment. Therefore, we suggest a two-step screening model: a brief validated tool plus an interview, rather than relying solely on a questionnaire. All patients should undergo a structured interview with a psychologist, who can perform case conceptualization [78] and determine individualized referral to evidence-based interventions such as Cognitive Behavioral Therapy [79], Mindfulness-Based interventions, [80] Meaning-Centered Therapy [81], or Dignity Therapy [82], depending on patient needs. It is also essential to consider the therapy format (individual vs. group; in-person vs. online), as these factors influence treatment efficacy, patient comfort, and practicality [6]. Evidence suggests that addressing psychological symptoms is not just an additional aspect of care but an essential component that can influence treatment outcomes, patient well-being, and overall quality of life.

Future research should include interventional trials to directly test the effectiveness of psychological support and targeted interventions in patients with pancreatic cancer. The burden of age-related symptoms may be an exciting area for further investigation. Longitudinal studies can explore how psychological symptoms evolve and change over time, considering treatment phases and post-treatment experiences, as the disease stage can greatly impact the outcomes.

The lack of consensus regarding the most effective screening tool represents a clinical, as well as a research problem. The use of questionnaires varies widely, and there is no established "gold standard" for identifying depression, anxiety, or distress across diverse cancer populations. Future research should support and prioritize large-scale, multinational studies to rigorously compare existing screening tools, determine which perform best in real-world clinical settings, or develop new, validated instruments that are brief, feasible, and sensitive to the needs of patients across different countries and healthcare

systems. Resolving this gap would improve patient care and enable more robust and comparable research on psychological interventions in cancer populations. Policymakers should support this research, as such funding would accelerate the generation of evidence needed to guide clinical practice and standardize psycho-oncology care globally. Besides, policymakers should ensure that every oncology ward employs a dedicated psycho-oncology team responsible for screening, case conceptualization, and delivery of these interventions, to ensure equitable access to psycho-oncological services.

## Conclusion

The high prevalence of psychological symptoms, including depression, anxiety, distress, fatigue, and sleep disturbances emphasizes the need for comprehensive psychological care as a core component of pancreatic cancer treatment.

## Supporting information

**S1 Documentum. Search Key.**
(PDF)

**S2 Documentum. Data synthesis – detailed methods.**
(PDF)

**S3 Documentum. Summary of all data underlying the findings described in the manuscript.** This table contains the complete dataset used to generate all results presented in the manuscript. 'Short'-within six months of cancer diagnosis; 'long'-beyond six months from cancer diagnosis. ROB- Risk of bias.
(PDF)

**S4 Documentum. List of excluded studies**.
(PDF)

**S1 Table. PRISMA checklist.** From: Page MJ, McKenzie JE, Bossuyt PM, Boutron I, Hoffmann TC, Mulrow CD, et al. The PRISMA 2020 statement: an updated guideline for reporting systematic reviews. BMJ 2021;372: n71. doi: 10.1136/bmj. n71.
(PDF)

**S2 Table. Basic characteristics of the included studies in the meta-analysis.** Basic characteristics of the included studies in the meta-analysis. BDI-Beck Depression Inventory; BSI-Brief Symptom Inventory; CIDI-Composite International Diagnostic Interview; DT- Distress Thermometer EORTC QLQ-FA12 -European Organization for Research and Treatment of Cancer Fatigue; ESAS-Edmonton Symptom Assessment System; ESASr-Edmonton Symptom Assessment System Revised; EQ-5D VAS- EuroQol-5D- visual analogue scale; FACIT-G-Functional Assessment of Cancer Therapy – General; HADS-Hospital Anxiety and Depression Scale; HIS-Hornheider Screening Instrument; ICD-International Classification of Diseases; MMPI-Minnesota Multiphasic Personality Inventory; NCCN (DT)-National Comprehensive Cancer Network's Distress Thermometer; PHQ-9-Personal Health Questionnaire; PO-Bado-Basic Documentation for Psycho-Oncology; PROMIS®-Patient-Reported Outcomes Measurement Information System; PSWQ-Penn State Worry Questionnaire; SAS-Self-rating Anxiety Scale; SCID-DSM-III-R-Structured Clinical Interview for Diagnostic and Statistical Manual of Mental Disorders; STAI-State Trait Anxiety Inventory. Time group: 'short'-within six months from the diagnosis of cancer; 'long'-beyond six months from the cancer diagnosis. NA-not applicable.
(PDF)

**S3 Table. Basic characteristic table of additional included articles that were not meta-analyzed.** Systematic review data table. BSI- Brief Symptom Inventory; PROMIS-Patient-Reported Outcomes Measurement Information System; NA-not applicable.
(PDF)

**S4 Table. The Joanna Briggs Assessment of the methodological quality.** N-no; Y-yes, U-unclear. The JBI appraisal checklist consists of 9 items and each item is assessed by scoring (yes = 1), (no = 0), and (unclear = 0).
(PDF)

**S1 Fig. Funnel plot analysis of depression.** The points represent the different studies. It shows the residuals on the x-axis against their corresponding standard errors. Egger's test p-value is 0.0413.
(PDF)

**S2 Fig. Funnel plot analysis of anxiety.** The points represent the different studies. It shows the residuals on the x-axis against their corresponding standard errors. Egger's test p-value is 0.7877.
(PDF)

**S3 Fig. Leave-one-out analysis of depression.** Figure shows how each individual study affects the overall estimate of the rest of the studies.
(PDF)

**S4 Fig. Leave-one-out analysis of the subset of depression.** Figure shows how each individual study affects the overall estimate of the rest of the studies.
(PDF)

**S5 Fig. Leave-one-out analysis of anxiety.** Figure shows how each individual study affects the overall estimate of the rest of the studies.
(PDF)

**S6 Fig. Leave-one-out analysis of distress.** Figure shows how each individual study affects the overall estimate of the rest of the studies.
(PDF)

**S7 Fig. Leave-one-out analysis of fatigue.** Figure shows how each individual study affects the overall estimate of the rest of the studies.
(PDF)

**S8 Fig. Leave-one-out analysis of sleep disturbances.** Figure shows how each individual study affects the overall estimate of the rest of the studies.
(PDF)

## Author contributions

**Conceptualization:** Péter Hegyi, Anna Sára Bognár, Dóra Andrea Sükös, Cristina Patoni, Eszter Ágnes Szalai, Nóra Hosszúfalusi, Bálint Mihály Erőss, Katalin Márta, Brigitta Teutsch.

**Data curation:** Dóra Andrea Sükös, Cristina Patoni, Nóra Hosszúfalusi, Bálint Mihály Erőss.

**Formal analysis:** Anna Sára Bognár, Dániel Sándor Veres.

**Funding acquisition:** Péter Hegyi, Eszter Ágnes Szalai, Brigitta Teutsch.

**Methodology:** Anna Sára Bognár, Eszter Ágnes Szalai.

**Project administration:** Anna Sára Bognár.

**Supervision:** Péter Hegyi, Eszter Ágnes Szalai, Katalin Márta, Brigitta Teutsch.

**Visualization:** Dániel Sándor Veres.

**Writing – original draft:** Anna Sára Bognár.

**Writing – review & editing:** Péter Hegyi, Dóra Andrea Sükös, Cristina Patoni, Dániel Sándor Veres, Eszter Ágnes Szalai, Nóra Hosszúfalusi, Bálint Mihály Erőss, Katalin Márta, Brigitta Teutsch.

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
