## [Decision Letter · Decision Letter 0]

15 Jul 2025

PONE-D-25-25313One out of four patients with pancreatic cancer experience psychological symptoms: a systematic review and meta-analysisPLOS ONE

Dear Dr. Hegyi,

Thank you for submitting your manuscript to PLOS ONE. After careful consideration, we feel that it has merit but does not fully meet PLOS ONE’s publication criteria as it currently stands. Therefore, we invite you to submit a revised version of the manuscript that addresses the points raised during the review process.

We look forward to receiving your revised manuscript.

Kind regards,

Cho Lee Wong, PhD

Academic Editor

PLOS ONE

Journal Requirements:

Additional Editor Comments:

Minor review according to the reviewer's comments. Moreover, please provide clear, high-quality images of the results.

Reviewers' comments:

Reviewer's Responses to Questions

**Comments to the Author**

1. Is the manuscript technically sound, and do the data support the conclusions?

Reviewer #1: Yes

Reviewer #2: Partly

2. Has the statistical analysis been performed appropriately and rigorously? 

Reviewer #1: Yes

Reviewer #2: Yes

3. Have the authors made all data underlying the findings in their manuscript fully available?

Reviewer #1: Yes

Reviewer #2: Yes

4. Is the manuscript presented in an intelligible fashion and written in standard English?

Reviewer #1: No

Reviewer #2: Yes

5. Review Comments to the Author

Reviewer #1: This systematic review and meta-analysis address a critical gap in psychology by quantifying psychological symptoms in pancreatic cancer (PC) patients. The study is methodologically rigorous (PRISMA-compliant, PROSPERO-registered) and clinically relevant. Below are specific commendations and suggestions for improvement

Reviewer #2: The study question is interesting and the methodology is reasonable.As mentioned in the manuscript,40–60% of European patients with cancer and their families face psychological distress that could be treated with adequate interventions. The high prevalence of psychological issues emphasizes the need for comprehensive psychological care as a core component of pancreatic cancer treatment. This study provides a potential direction for future psychological interventions for pancreatic cancer patients and their families.

6. PLOS authors have the option to publish the peer review history of their article (what does this mean?). If published, this will include your full peer review and any attached files.

Reviewer #1: **Yes:** Gurmessa Dessale Assege

Reviewer #2: No

---

## [Author Response · Author response to Decision Letter 1]

27 Aug 2025

Dear Editor,

Dr Cho Lee Wong,

Thank you for giving us the opportunity to submit a revised draft of our manuscript titled One out of four patients with pancreatic cancer experience psychological symptoms: a systematic review and meta-analysis to PLOS ONE. We appreciate the time and effort that you and the reviewers have dedicated to providing your valuable feedback on our manuscript. We are grateful to the reviewers for their insightful comments on our paper. We have been able to incorporate changes to reflect most of the suggestions provided by the reviewers. We uploaded a marked-up copy of the manuscript that highlights changes made to the original version, labeled as 'Revised Manuscript with Track Changes'.

As well as, an unmarked version of the revised paper without tracked changes, labeled as ‘'Manuscript'.

Here is a point-by-point response to the reviewers’ comments and concerns.

Again, thank you for your very valuable time invested in our manuscript.

Best regards,

Péter Hegyi and his team.

Dear Reviewers,

Thank you very much for your valuable feedback. We greatly appreciated your comments which have significantly improved our manuscript. Please find below our point-by-point responses. As all comments appeared under “By Author,” we were unsure which comment belonged to which reviewer. Therefore, we have addressed the comments in the order in which they appeared.

We hope that our revised manuscript will meet your expectations and you will find it suitable for publication. Please let us know if any further changes are required.

Again, thank you for your very valuable time invested in our manuscript.

Best regards,

Péter Hegyi and his team.

● Comment 1: “Many systematic reviews there about Psychological stress and pancreatic cancer patients but more interested if revies foucosed on why one patients expericed from the four patients”

Response 1: Thank you for your comment. We would like to clarify that the goal of our systematic review and meta-analysis was to estimate the prevalence of psychological symptoms among patients with pancreatic cancer, rather than to identify individual-level predictors of psychological distress. The title “One out of four patients with pancreatic cancer experience psychological symptoms” was formulated to reflect our key findings. We agree that exploring the underlying factors that determine why certain patients experience psychological symptoms is an important question, but it was beyond the scope of this review.

Action 1: Not needed.

● Comment 2: “Try to avoid an abbreviation from the abstract.”

Response 2: Thank you for your suggestion — we fully agree. In the original abstract, we used two abbreviations: PC (pancreatic cancer), which appeared three times, and CI (confidence interval), which was used six times. We have now removed the abbreviation PC and replaced it with the full term. Given the frequent use of CI, and its widespread recognition in academic writing, we have retained it to ensure clarity and readability.

Action 2: We removed the abbreviation PC from the abstract and replaced it with the full term pancreatic cancer. We retained CI (confidence interval) due to its frequent use and common understanding in scientific literature. *We left the PC (pancreatic cancer) abbreviation in the text (apart from the abstract) as it appears 22 times, in order to improve the readability and flow of the manuscript.

● Comment 3: “Background (first Paragraph) The first sentence is factual but could better link to the study's focus on psychological burden. Try to revision”

Response 3: Thank you for your valuable comment regarding the first paragraph of the abstract. We appreciate your suggestion to strengthen the link between the high mortality of pancreatic cancer and the psychological burden.

Action 3:Therefore, we added the following sentence to strengthen the link:

“As the prognosis remains extremely poor, this trend brings not only a growing clinical challenge, but also a significant psychological toll for patients.”

● Comment 3: “Try to revise”

Response 3: Thank you for your suggestion. To fully understand the request could you detail it further? Thank you in advance.

Action 3: None.

● Comment 4:” I think this parts methodology parts so if you add like:

1. The use data extracted and exported must mandatory

2. Also what you search articles many of are you there is no others relevant studies from others searching database what you are (MEDLINE, Cochrane, and Embase) please accurately try add realy get those data reality 1.

3. Add the risk of bias was assessed using a methods test

4. Also the later results present by what? add”

Response 4:

1. If we understand your comment correctly, you suggest to explicitly state that data was extracted and exported for analysis? If so, we have clarified in the abstract that data were extracted and exported for synthesis. We hope this improves the methodological transparency of our manuscript.

2. Could you please assure us that your comment means to ensure that the literature search was comprehensive enough and included all relevant studies? The PRISMA guidelines recommend searching in two databases, however, in order to find all the relevant studies, we conducted our search in more than two databases — MEDLINE, Embase, and Cochrane — the three biggest and mostly accepted databases in systematic reviews. We believe this approach ensures sufficient coverage and minimizes the risk of missing relevant literature.

3. and 4. Thank you for your comment. We would kindly ask for clarification, as we are not entirely certain we have understood your suggestion and question correctly. If your comment refers to the need for more detail regarding the risk of bias assessment and the way results were presented, we would like to note that the journal imposes a strict word limit for the abstract. Therefore, we aimed to balance methodological transparency with conciseness. If further clarification is required in the abstract, we could add a sentence such as:

“Risk of bias was assessed using the Joanna Briggs Institute Critical Appraisal Tool by two independent reviewers, with disagreements resolved by a third. A frequentist random-effects meta-analysis was performed, and results are presented with forest plots, heterogeneity statistics, and funnel plots for publication bias.”

However, we believe that adding this 44-word sentence would exceed the abstract’s length unnecessarily and would require omitting other key content.

Action 4: 1. The following sentence was added: “Relevant data were extracted and exported for quantitative synthesis.”

2., 3., 4.: None.

● Comment 5: “Make clarify please

1. Double Percentage Problem: When you write "27%; CI= [0.18-0.39%]", you are First converting the proportion to 27% (correct). But then showing the CI bounds as percentages of a percentage (0.18% of 27%), which is wrong. Generally for all results write: Correct Formats (Choose One): Proportion Format (Preferred for Meta-Analyses): 0.27 (95% CI: 0.18-0.39) Most statistically accurate, Avoids percentage confusion and Common in epidemiology literature. 2. Percentage Format (If Required): 27% (95% CI: 18%-39%) More intuitive for clinicians, Must multiply decimals by 100 and Keep "%" on both estimate and CI bounds. Correct interpretation If your analysis gave a proportion of 0.27 (27%) with CI [0.18-0.39], these decimals are already proportion”

Response 5: Thank you for highlighting this. We fully agree with your suggestion.

Action 5: We now made the correction and used the following format: 0.27 (95% CI: 0.18-0.39)

● Comment 6: “Why? Combined two sentences into one cohesive thought. Removed redundant phrasing (most and least effective... beneficial approaches).????????”

Response 6: We understand your suggestion to improve clarity and reduce redundancy in the final sentences of the abstract.

Action 6: In response, we have merged the two sentences into a more cohesive conclusion and recommendation, in line with standard expectations for systematic reviews, see next comment (Comment 7).

● Comment 7: “Make sure the contents have conclusion and recommendation format of systematic review?”

Response 7: Thank you for the recommendation.

Action 7: We have now a better conclusion and recommendation, in line with standard expectations for systematic reviews. The revised section now reads:

“These findings highlight the urgent need to integrate psychological care into pancreatic cancer treatment. Future research should explore sources of heterogeneity to better identify which approaches provide the most psychological benefit and tailor interventions to patient needs accordingly.”

● Comment 8: “Make consistent the journals protocol point is before reference or after reference?”

Response 8: Thank you for highlighting this out.

Action 8: We corrected the mistake accordingly.

● Comment 9: “Re write this protocol registration”

Response 9: Thank you for your comment. We would kindly ask for clarification, as the current phrasing appears to be grammatically correct and clearly communicates both our adherence to PRISMA and the PROSPERO registration of the protocol. We would be happy to revise the sentence if you could specify what aspect of the wording you found unclear or problematic.

Action 9: None.

● Comment 10: “Use reference protocol registration on international prospective of systematic review. Also if changes like authors due to different case elaborate in this”

Response 10: Thank you for your suggestion. We assume you are referring to the phrasing of the protocol registration. Our original sentence was grammatically correct and conveyed the essential information, but we are happy to revise it slightly.

Action 10: We included the full name of PROSPERO as “The International Prospective Register of Systematic Reviews” for clarity. We also confirm that no deviations (including authorship or content changes) occurred from the registered protocol.

● Comment 11: “My be for me weak because why not elaborate others from others journal”

Response 11: Thank you for your feedback. We are not entirely sure we understood your concern correctly. If you are suggesting that our search strategy should have included more databases or sources beyond MEDLINE, Embase, and Cochrane, we would like to clarify that we deliberately focused on these three large and widely used databases in line with PRISMA recommendations, which require a minimum of two databases.

Action 11: None.

● Comment 12: “Why only focus only 3 journal for collects article 23,282? That mean there is no relevant study others journal? Example I get from Google, AJOL, Google Scholar, and Hinari and others”

Response 12: We chose MEDLINE, Embase, and Cochrane CENTRAL because these are among the most comprehensive and widely used biomedical databases and are standard in systematic reviews following PRISMA guidelines. Together, they cover a broad spectrum of high-quality peer-reviewed literature, including studies from a wide range of journals and countries. We believe this approach strikes a reasonable balance between breadth and methodological quality.

Action 12: None.

● Comment 13: ”Justification why bounded between this before and afters?”

Response 13: Thank you for your comment. We believe there may have been a misunderstanding: our initial systematic search was conducted on May 22, 2022, covering all available literature up to that point, regardless of publication year. Due to a delay in submission, we performed an updated search on March 3, 2025, to ensure the inclusion of the most recent studies. The search strategy was not limited to publications between these two dates — rather, the March 2025 search served to supplement the original data.

Action 13: None.

● Comment 14: “Add General characteristics of studies included in the meta-analysis”

Response 14: Thank you for your comment. We would like to note that the general characteristics of the studies included in the meta-analysis are summarized in the main text and presented in detail in the Supporting Information (Table S2). The text includes the number of studies addressing each psychological domain and the range of participant numbers. Additional studies included in the systematic review are detailed in Table S3.

As these tables are self-explanatory and clearly referenced in the manuscript, we believe they provide sufficient background without unnecessarily extending the main text. However, if there are specific details you find missing or unclear, we would be grateful for further clarification.

Action 14: None.

● Comment 15: “Clearly write the quality score of the study with draws tables”

Response 15: Thank you for your comment. We would like to respectfully note that the methodological quality assessment is already mentioned in the manuscript, and the table in the Supplementary information was mentioned (Table S4) too. However, we understand that it might be concise.

Action 15: We rewrote the section: “The quality of the included studies was assessed using the Joanna Briggs Institute Critical Appraisal Tool. Detailed scores and assessment outcomes are provided in Supporting Information, Table S4. Studies were grouped by risk of bias (high, medium, low), and these groupings are also reflected in the forest plots.” We believe this presentation is clear and appropriately detailed for the reader.

● Comment 16: “The same above comments unders abstract of results Double Percentage Problem”

Response 16: Thank you for highlighting this. We fully agree with your suggestion.

Action 16: We made the correction accordingly.

● Comment 17: “Add sensitivity analysis. Clearly”

● Response 17: Thank you for your comment. We acknowledge that the Leave-one-out analysis is a well-established sensitivity analysis method. While the manuscript reports the Leave-one-out results, we agree that explicitly labeling it as a sensitivity analysis would improve clarity.

● Action 17: We added the term sensitivity analysis to the heading: “Publication Bias, Heterogeneity, and Sensitivity Analyses”

● Comment 18: “Add Author’s contribution: Abbreviation and acronyms (optional) but I see in abstract like PC, PRISMA”

● Response 18: Thank you for your comment. The Author Contributions section was completed as part of the submission process on the platform. Regarding abbreviations and acronyms, we have used only a few, all of which are clearly defined in the manuscript. For example, “PC” is defined on first use. We also note that “PRISMA” does not appear in the abstract, so no abbreviation for it is included there.

● Action 18: None needed

Additional clarifications

● Comment 19:

“General Strengths:

The study addresses an important gap in psycho-oncology literature with clear clinical relevance.

Comprehensive methodology following PRISMA guidelines with PROSPERO registration

Large sample sizes across multiple psychological domains (n=78,930 for depression)

Rigorous statistical approach accounting for heterogeneity and bias”

● Response 19: Thank you for these encouraging comments and greatly appreciate the recognition of our study’s strengths.

● Action 19: Not needed.

● Comment 20:

“Abstract:

The abstract would benefit from specifying the exact databases searched and inclusion dates

Consider adding the Joanna Briggs Institute tool as your quality assessment method

The conclusion could be strengthened by mentioning specific intervention recommendations”

● Response 20: Thank you for these helpful suggestions. Due to the word count limit, we were unable to add all the suggested information and prioritized including details regarding the study’s quality assessment. Regarding the suggestion to include intervention recommendations, we respectfully note that, due to the word count limit and the lack of clearly defined, evidence-based interventions, we did not add this to the abstract. Indeed, one of the central points of our study is precisely the need for future research to identify and evaluate effective interventions for this highly burdened patient group.

● Action 20: We added the following sentence to the abstract: the “Study quality was assessed using the Joanna Briggs Institute tool.”

● Comment 21:

“I

---

## [Decision Letter · Decision Letter 1]

6 Oct 2025

PONE-D-25-25313R1One out of four patients with pancreatic cancer experience psychological symptoms: a systematic review and meta-analysisPLOS ONE

Dear Dr. Hegyi,

Thank you for submitting your manuscript to PLOS ONE. After careful consideration, we feel that it has merit but does not fully meet PLOS ONE’s publication criteria as it currently stands. Therefore, we invite you to submit a revised version of the manuscript that addresses the points raised during the review process.

We look forward to receiving your revised manuscript.

Kind regards,

Cho Lee Wong, PhD

Academic Editor

PLOS ONE

Journal Requirements:

Additional Editor Comments:

Minor revision is needed.

Reviewers' comments:

Reviewer's Responses to Questions

**Comments to the Author**

1. If the authors have adequately addressed your comments raised in a previous round of review and you feel that this manuscript is now acceptable for publication, you may indicate that here to bypass the “Comments to the Author” section, enter your conflict of interest statement in the “Confidential to Editor” section, and submit your "Accept" recommendation.

Reviewer #3: (No Response)

Reviewer #4: All comments have been addressed

Reviewer #5: All comments have been addressed

Reviewer #6: All comments have been addressed

Reviewer #7: (No Response)

Reviewer #8: (No Response)

2. Is the manuscript technically sound, and do the data support the conclusions?

Reviewer #3: Partly

Reviewer #4: Yes

Reviewer #5: Yes

Reviewer #6: Yes

Reviewer #7: Yes

Reviewer #8: Yes

3. Has the statistical analysis been performed appropriately and rigorously? 

Reviewer #3: No

Reviewer #4: Yes

Reviewer #5: Yes

Reviewer #6: Yes

Reviewer #7: I Don't Know

Reviewer #8: Yes

4. Have the authors made all data underlying the findings in their manuscript fully available?

Reviewer #3: Yes

Reviewer #4: Yes

Reviewer #5: Yes

Reviewer #6: Yes

Reviewer #7: Yes

Reviewer #8: Yes

5. Is the manuscript presented in an intelligible fashion and written in standard English?

Reviewer #3: Yes

Reviewer #4: Yes

Reviewer #5: Yes

Reviewer #6: Yes

Reviewer #7: Yes

Reviewer #8: Yes

6. Review Comments to the Author

Reviewer #3: Below, I outline my major concerns and suggestions for revision.

Major Comments:

1. Statistical Presentation of Proportions and Confidence Intervals

Although the authors have corrected the format of confidence intervals in the abstract (e.g., from “27%; CI=[0.18–0.39%]” to “0.27 (95% CI: 0.18–0.39)”), this correction has not been consistently applied throughout the manuscript. For instance, in the Results section (pp. 25–27), the same error persists (e.g., “proportion was 27%; [95% CI: CI=[0.18-0.39]%”).

o Recommendation: Please ensure that all prevalence estimates and confidence intervals are reported consistently in the proportion format (e.g., 0.27, 95% CI: 0.18–0.39) rather than mixing percentages and proportions. This is essential for statistical accuracy and reader comprehension.

2. Clarity and Completeness of the Methods Section

o The authors mention the use of the Joanna Briggs Institute (JBI) tool for quality assessment but do not provide a summary or interpretation of the results in the main text. The reader is referred to Supplementary Table S4, but a brief summary of the overall quality of included studies (e.g., how many were low, moderate, or high risk) should be included in the main manuscript to enhance transparency.

o The term “frequentist random-effects models” is used, but the rationale for choosing a frequentist over a Bayesian approach is not discussed. A brief justification would strengthen the methodology.

3. Heterogeneity and Sensitivity Analyses

o The I² values are exceptionally high (e.g., 98.2% for depression), indicating substantial heterogeneity. The authors briefly mention heterogeneity in the Discussion but do not sufficiently explore its potential sources (e.g., differences in measurement tools, cultural contexts, study populations).

o Recommendation: A more in-depth discussion of the possible reasons for heterogeneity is needed. Consider conducting subgroup or meta-regression analyses (if feasible) to explore sources of variation, such as geographic region, assessment tool, or time since diagnosis.

4. Clinical and Policy Implications

o The authors appropriately call for integrating psychological care into pancreatic cancer treatment. However, the recommendations remain general.

o Recommendation: Provide more specific, actionable suggestions for clinicians and policymakers. For example:

Routine screening for psychological symptoms using validated tools.

Referral pathways to mental health services.

Training for oncology care teams in basic psychological support.

5. Limitations Section

o The limitations are mentioned briefly but should be expanded. Key issues include:

High heterogeneity among studies.

Potential underreporting of psychological symptoms due to stigma.

Overrepresentation of certain geographic regions (e.g., Europe and North America), limiting generalizability.

Variability in measurement tools and diagnostic thresholds.

6. Data Availability Statement

o The current statement, “Data available in original articles,” is not sufficient under PLOS ONE’s data policy.

o Recommendation: Please revise to state clearly that all data extracted for the meta-analysis are included in the manuscript and/or Supporting Information files. If data are only available from original studies, this must be explicitly stated with reasons.

Minor Comments:

• Abstract: The sentence “Study quality was assessed using the Joanna Briggs Institute tool” is appropriate but should specify that this was done for included studies.

• Figures: Ensure that all forest plots and figures are clearly labeled and that prediction intervals are included where appropriate.

• References: Check consistency in reference formatting throughout the manuscript.

Reviewer #4: Thank you for giving us the opportunity to review our manuscript titled One out of four patients with pancreatic cancer experience psychological symptoms: a systematic review and meta-analysis to PLOS ONE. Although manuscript is good and well structure, but there are some comments to help strengthen your article.

comment (1):the authors should discuss more critically how differences in measurement tools may have influenced prevalence estimates.

comment(2): timing of symptoms onset ,the study mentions " short vs. long" time from diagnosis but concludes timing does not affect outcomes. please clarify

comment (3): the recommendation remain general not dependent on conclusion that called for psychological care with PC treatment.

comment (4) in line 181: section "Basic characteristics of studies included in the meta-analysis." please clarify why the 2 extra studies not included in mete-analysis.

comment (5): mention please how many studies for each categories( low, moderate, & sever ) in all symptoms.

Reviewer #5: The manuscript is well-executed and addresses a critical issue in the management of pancreatic cancer patients. No revisions are needed, and the revisions made are thoughtful and improve the clarity and transparency of the study.

Therefore, I recommend accepting the manuscript without revision.

Reviewer #6: (No Response)

Reviewer #7: Thank you for the opportunity to review this interesting paper. The topic is highly relevant and contributes to the growing body of knowledge on oncology. The manuscript is generally well-structured and clearly written. However, few comments I provided you may consider;

1. Title: The title reads a bit like a research finding rather than a formal title.

I suggest changing it to something like; “Psychological Symptoms in Pancreatic Cancer: Prevalence and Evidence from a Systematic Review and Meta-Analysis”

2. In line 219, under the Distress subtitle, you mentioned “anxiety.” I believe you intended to write “distress” instead. Kindly check and revise for accuracy.

3. Early in the discussion, you repeat prevalence percentages already mentioned in the Results (lines 260–261, 267). Instead of re-listing numbers, focus on interpretation (“Our findings confirm that anxiety and depression are particularly common, with rates similar to or exceeding other cancers.”).

4. Be precise about definitions. Sometimes “distress” is used interchangeably with “anxiety,” which may confuse readers. Define how distress was measured in your included studies.

5. In line 347, instead of “as we know it today,” use more formal phrasing: “To the best of our knowledge, this is the first …”

6. Under implications, be more specific about recommended practice: e.g., recommend routine screening for depression/anxiety/distress using validated tools (like HADS, Distress Thermometer).

7. For policy implications, suggest strategies: e.g., inclusion of psycho-oncology in national cancer care guidelines, training oncology staff in psychological screening.

8. For research implications, highlight the need for interventional trials more to test effectiveness of psychological support in PC patients.

9. In line 382, consider tightening wording: “psychological alterations” → “psychological symptoms” (for more consistency with earlier sections).

Reviewer #8: The manuscript is well-structured, written in clear academic English, and follows a logical scientific format (Introduction, Methods, Results, Discussion, Conclusion). Figures and tables are relevant, and the authors provide supplementary data as required. However, there are areas that should be improved:

1. Update of Statistics – In the Introduction, the prevalence and mortality numbers for pancreatic cancer are based on 2023 reports. Since this is a 2025 submission, the authors should cite the most recent epidemiological data using Siegel 2025 Cancer Statistics to ensure accuracy

2. Link to Survival Outcomes – The Introduction highlights the psychological burden but does not explicitly connect these symptoms to clinical outcomes. The authors should mention that depression, anxiety, and distress can negatively affect treatment adherence, quality of life, and survival rates in pancreatic cancer patients. Including this would strengthen the rationale.

3. Figures/Tables – Figure 1 (PRISMA diagram) should be redrawn in higher resolution. Some table footnotes should be expanded to explain abbreviations and statistical notations more clearly.

4. Measurement Tools – Multiple instruments were used across studies (e.g., HADS, PHQ-9, BDI). A comparative table showing cut-off scores and interpretability would help clarify variability across the tools.

5. Publication Bias – Funnel plots were only presented for depression and anxiety. The authors should clearly acknowledge the limitation that other outcomes could not be assessed due to smaller sample sizes.

6. Clinical Implications – The conclusion currently recommends “targeted interventions,” but this is too broad. The authors should propose more concrete strategies such as: Routine psychological screening at diagnosis, or Integration of psycho-oncology referrals,

Overall Recommendation

This manuscript addresses an important knowledge gap and contributes to the literature on psychological burden in pancreatic cancer. With revisions — particularly updating statistics, linking psychological symptoms to survival, and making recommendations more actionable — the paper will provide stronger clinical implications.

7. PLOS authors have the option to publish the peer review history of their article (what does this mean?). If published, this will include your full peer review and any attached files.

Reviewer #3: No

Reviewer #4: No

Reviewer #5: **Yes:** Mohammed Al Bazroun

Reviewer #6: No

Reviewer #7: No

Reviewer #8: No

---

## [Author Response · Author response to Decision Letter 2]

21 Nov 2025

Dear Editors,

Dr Cho Lee Wong,

Thank you for giving us the opportunity to submit a revised draft of our manuscript titled One out of four patients with pancreatic cancer experience psychological symptoms: a systematic review and meta-analysis to PLOS ONE. We appreciate the time and effort that you and the reviewers have dedicated to providing your valuable feedback on our manuscript. We are grateful to the reviewers for their insightful comments on our paper. We have been able to incorporate changes to reflect most of the suggestions provided by the reviewers.

We uploaded a marked-up copy of the manuscript that highlights changes made to the original version, labeled as 'Revised Manuscript with Track Changes'. As well as an unmarked version of the revised paper without tracked changes, labeled as ‘'Manuscript'.

In accordance with the editor’s request, we have reviewed and updated the reference list. Several references were revised or replaced in the Introduction to reflect the most up-to-date statistical data, and additional references were added in the Discussion section to support the revisions made in response to the reviewers’ comments. We confirm that no retracted articles are cited in the revised manuscript.

Here is a point-by-point response to the reviewers’ comments and concerns.

Again, thank you for your very valuable time invested in our manuscript.

Best regards,

Péter Hegyi and his team.

Dear Reviewers,

Thank you very much for your valuable feedback. We greatly appreciated your comments which have significantly improved our manuscript. Please find below our point-by-point responses.

We hope that our revised manuscript will meet your expectations, and you will find it suitable for publication. Please let us know if any further changes are required.

Again, thank you for your very valuable time invested in our manuscript.

Best regards,

Péter Hegyi and his team.

Reviewer #3:

Comment 1: “1. Statistical Presentation of Proportions and Confidence Intervals

Although the authors have corrected the format of confidence intervals in the abstract (e.g., from “27%; CI=[0.18–0.39%]” to “0.27 (95% CI: 0.18–0.39)”), this correction has not been consistently applied throughout the manuscript. For instance, in the Results section (pp. 25–27), the same error persists (e.g., “proportion was 27%; [95% CI: CI=[0.18-0.39]%”).

Recommendation: Please ensure that all prevalence estimates and confidence intervals are reported consistently in the proportion format (e.g., 0.27, 95% CI: 0.18–0.39) rather than mixing percentages and proportions. This is essential for statistical accuracy and reader comprehension.”

Response 1: Thank you for this helpful observation. We agree that consistency in the statistical presentation of proportions and confidence intervals is essential for accuracy and readability.

Action 1: Revised all instances where prevalence values were expressed as percentages (e.g., “27%”) or mixed formats (e.g., “27%; 95% CI: 0.18–0.39%”) to the standardized format: 0.27 (95% CI: 0.18–0.39).

Comment 2: “2. Clarity and Completeness of the Methods Section

2.1. The authors mention the use of the Joanna Briggs Institute (JBI) tool for quality assessment but do not provide a summary or interpretation of the results in the main text. The reader is referred to Supplementary Table S4, but a brief summary of the overall quality of included studies (e.g., how many were low, moderate, or high risk) should be included in the main manuscript to enhance transparency.”

Response 2.1: Thank you for the suggestion. We agree that including a concise summary of the JBI quality assessment results in the main text improves the clarity and transparency of the Results sections.

Action 2.1: We have added the following sentence summarizing the overall quality of the included studies in the Results section, under “Assessment of methodological quality”. The paragraph now reads as follows:

“Assessment of methodological quality for evident synthesis

The quality of the included studies was assessed using the Joanna Briggs Institute Critical Appraisal Tool. Studies were grouped by risk of bias (high, medium, low), and these groupings are also reflected in the forest plots. Among the 28 included studies, 21 were rated as low risk of bias, six as moderate risk, and one as high risk according to the JBI tool. Detailed scores and assessment outcomes are provided in Supporting Information, Table S4.”

Comment 2.2: “2. Clarity and Completeness of the Methods Section

The term “frequentist random-effects models” is used, but the rationale for choosing a frequentist over a Bayesian approach is not discussed. A brief justification would strengthen the methodology.”

Response 2.2: We appreciate the reviewer’s thoughtful comment. While both Bayesian and frequentist frameworks are suitable for random-effects meta-analysis, we believe that the choice between them does not substantially affect the conclusions in this context. We therefore adopted the frequentist approach, which is the conventional and most commonly reported method in the literature. Since the vast majority of studies implicitly use this framework without explicitly stating it, we feel that a detailed discussion of the rationale in the Methods section would not substantially enhance the clarity or reproducibility of our analysis.

Action 2.2: None.

Comment 3: “3. Heterogeneity and Sensitivity Analyses

The I² values are exceptionally high (e.g., 98.2% for depression), indicating substantial heterogeneity. The authors briefly mention heterogeneity in the Discussion but do not sufficiently explore its potential sources (e.g., differences in measurement tools, cultural contexts, study populations).

Recommendation: A more in-depth discussion of the possible reasons for heterogeneity is needed. Consider conducting subgroup or meta-regression analyses (if feasible) to explore sources of variation, such as geographic region, assessment tool, or time since diagnosis.”

Response 3: Thank you for this important observation. We also acknowledged that the I² values were exceptionally high across all examined outcomes, therefore we put a great effort on identifying key sources of heterogeneity as follows:

Population Diversity, we wrote: "The heterogeneity observed in the meta-analysis suggests that patient populations are diverse across studies. Differences in demographics, disease stages, treatment modalities, and geographic locations can impact reported values."

Cultural Factors, we wrote: "A more homogeneous population can provide a more accurate value for prevalence, as cultural and societal factors influence how individuals express and perceive psychological symptoms, as seen in the studies included with high or low prevalence, e.g., in the study by Akizuki et al., Variations in cultural norms, attitudes toward mental health, and the availability of support services can contribute to differences in reported proportions across studies."

Assessment tool, we wrote: "The measurement tools included in the studies are widely used in clinical and research settings ... Although they all offer standardized measures... they also raise questions of comparability and might affect the results." We further elaborated: "Variations in administration protocols, scoring methods, cultural adaptations, and severity across different studies can further complicate the comparability and interpretability of findings from these instruments."

We understand the need to further highlight the high heterogeneity observed across studies. We further made the existing points more direct and prominent to explain the sources of high heterogeneity.

We also agree that further exploration of heterogeneity would be beneficial. However, the available data were insufficient to perform subgroup or meta-regression analyses in a reliable and reliable way.

Action 3: We added: “This analysis also confirms that the high observed heterogeneity might be multifactorial and not caused by a single influential outlier study”. The discussion of heterogeneity has been substantially expanded between lines 309–362 (in the tracked-changes version of the manuscript). Additionally, we have revised the Limitations section to highlight the high heterogeneity observed across studies.

Comment 4: ” Clinical and Policy Implications - The authors appropriately call for integrating psychological care into pancreatic cancer treatment. However, the recommendations remain general.

Recommendation: Provide more specific, actionable suggestions for clinicians and policymakers. For example: Routine screening for psychological symptoms using validated tools. Referral pathways to mental health services. Training for oncology care teams in basic psychological support.”

Response 4: Thank you for this constructive suggestion. We agree that providing more specific, actionable recommendations strengthens the clinical and policy implications of our findings.

Action 4: We have revised the Implications for practice, research and policymakers’ section to include more concrete strategies. Line 406-462 in the Manuscript with Tracked Changes file.

Comment 5: “Limitations Section. The limitations are mentioned briefly but should be expanded. Key issues include: High heterogeneity among studies. Potential underreporting of psychological symptoms due to stigma. Overrepresentation of certain geographic regions (e.g., Europe and North America), limiting generalizability. Variability in measurement tools and diagnostic thresholds”

Response 5: Thank you for your highlightings. We fully agree with your suggestion.

Action 5: We added to the Limitation section the following:

“ [...]These factors, together with cultural or language adaptations, differences in patient populations contribute to the high heterogeneity observed across analyses (I² consistently >90%), which should be considered when interpreting pooled prevalence estimates.”

We believe further repetition is unnecessary, as these factors are already discussed in detail in the Discussion section.

Comment 6: “Data Availability Statement. The current statement, “Data available in original articles,” is not sufficient under PLOS ONE’s data policy.

Recommendation: Please revise to state clearly that all data extracted for the meta-analysis are included in the manuscript and/or Supporting Information files. If data are only available from original studies, this must be explicitly stated with reasons.”

Response 6: We thank you for pointing this out. We agree that our original Data Availability Statement did not fully align with PLOS ONE’s policy and have revised it accordingly for clarity and transparency.

Action 6: We have updated the Data Availability Statement to the following: “All data extracted for the meta-analysis are included in the manuscript tables, figures and Supporting Information files. A summarized excel table of all data is available for publicity upon request with reasons.”

Comment 7: “Abstract: The sentence “Study quality was assessed using the Joanna Briggs Institute tool” is appropriate but should specify that this was done for included studies.”

Response 7: Thank you for the recommendation.

Action 7: We have revised the sentence to read: “Study quality for the included studies was assessed using the Joanna Briggs Institute tool.”

Comment 8: “Figures: Ensure that all forest plots and figures are clearly labeled and that prediction intervals are included where appropriate.”

Response 8: Upon careful review, all figures and captions appear to us to be clearly labeled. However, we would appreciate further clarification regarding which specific aspects were found unclear. With a bit more precision, we would be glad to make any additional adjustments or clarifications to ensure the figures fully meet the expectations.

Regarding confidence intervals, Prediction intervals are already displayed in all figures where available, and this is explicitly noted in our manuscript: “We summarized the findings from the meta-analysis in forest plots. We also reported the prediction intervals (i.e., the expected range of effects of future studies) of results, where applicable: the number of studies was large enough (>5), and the effects were not too heterogeneous.”

Action 8: None.

Comment 9: “References: Check consistency in reference formatting throughout the manuscript.”

Response 9: Thank you for pointing this out. We have carefully reviewed all references to ensure full consistency as authors - references with more than six authors contain “et al.”, - title – Journal – year – additions

Action 9: All references have been rechecked and standardized.

Reviewer #4:

Comment 1: “The authors should discuss more critically how differences in measurement tools may have influenced prevalence estimates.”

Response 1: Thank you for this valuable comment. We have updated the Discussion section to provide a more critical reflection, including how differences in measurement tools and cut-off thresholds may have influenced the reported prevalence estimates.

Action 1: The Discussion has been revised accordingly (lines 309–362 in the Revised Manuscript with Track Changes), and we hope that these additions meet your expectations.

Comment 2: “Timing of symptoms onset, the study mentions " short vs. long" time from diagnosis but concludes timing does not affect outcomes. Please clarify.”

Response 2: We thank you for this helpful comment and appreciate the opportunity to clarify this point. The “short” versus “long” categorization refers to the time elapsed since pancreatic cancer diagnosis, where “short” indicates within six months and “long” refers to beyond six months. These groupings were applied based on the data available in the included studies and are displayed in the forest plots for visual comparison. However, a formal statistical subgroup analysis was not conducted for this variable, as the number of studies providing time-specific data was limited and did not allow for a reliable quantitative comparison. Our conclusion that timing did not appear to affect outcomes was therefore based on visual inspection of the forest plots, which showed similar prevalence estimates between the “short” and “long” categories.

Action 2: None.

Comment 3: ”The recommendation remain general not dependent on conclusion that called for psychological care with PC treatment.”

Response 3: We have revised the Implications for Practice, Research, and Policymakers section to ensure that our recommendations are more specific and clearly aligned with the conclusion calling for the integration of psychological care into pancreatic cancer treatment.

Action 3: These changes have been made starting from line 406 in the Revised Manuscript with Track Changes.

Comment 4: “"Basic characteristics of studies included in the meta-analysis." please clarify why the 2 extra studies not included in mete-analysis.”

Response 4: Thank you for your comment. The reasons for excluding the three studies from the quantitative synthesis are presented in the Supplementary Materials (Table S3), which includes a dedicated “Reason for exclusion from meta-analysis” column. This table provides a comprehensive yet quick overview, allowing readers to access all relevant information efficiently. We believe this is the most transparent and practical way to present these details.

Action 4: No changes have been made, as the information is already clearly and comprehensively presented in Supplementary Table S3. If you think that it is necessary. We would of course add an additional brief note in the main text.

Comment 5: “Mention please how many studies for each categories( low, moderate, & sever ) in all symptoms.”

Response 5: We thank you for this helpful suggestion. Data on severity were available only for depression, and we have now specified the number of studies contributing to each severity subgroup. For the other psychological symptoms (anxiety, distress, fatigue, and sleep disturbances), we didn’t have enough data, and therefore subgroup analysis by severity was not feasible.

Action 5: We rewrote the section: “In the studies where data based on severity was available the subset analysis based on severity showed that the proportion of mild de

---

## [Editor Report · Decision Letter 2]

9 Dec 2025

PONE-D-25-25313R2One out of four patients with pancreatic cancer experience psychological symptoms: a systematic review and meta-analysisPLOS One

Dear Dr. Hegyi,

Thank you for submitting your manuscript to PLOS ONE. After careful consideration, we feel that it has merit but does not fully meet PLOS ONE’s publication criteria as it currently stands. Therefore, we invite you to submit a revised version of the manuscript that addresses the points raised during the review process.

We look forward to receiving your revised manuscript.

Kind regards,

Cho Lee Wong, PhD

Academic Editor

PLOS One

Journal Requirements:

Additional Editor Comments:

Suggested a minor revision.

---

## [Author Response · Author response to Decision Letter 3]

12 Feb 2026

Dear Dr Cho Lee Wong,

Thank you for the opportunity to submit a revised version of our manuscript. One out of four patients with pancreatic cancer experience psychological symptoms: a systematic review and meta-analysis to PLOS ONE. We sincerely appreciate the time and effort you and the reviewers have dedicated to providing constructive feedback. We have carefully considered all of your comments and incorporated revisions to address the suggestions provided.

We uploaded a marked-up copy of the manuscript that highlights changes made to the original version, labeled as 'Revised Manuscript with Track Changes'.

As well as, an unmarked version of the revised paper without tracked changes, labeled as ‘'Manuscript'.

Here is a point-by-point response to the reviewer’s comments and concerns.

Again, thank you for your very valuable time invested in our manuscript.

Best regards,

Péter Hegyi and his team.

Comment 1: “ Abstract. Interpretation of pooled estimates: Given the very high heterogeneity (I² >90%), the Abstract should be more cautious in presenting point estimates without qualification.”

Response 1: We thank you for this suggestion and agree that the high level of heterogeneity requires cautious interpretation. However, due to the PLOS ONE abstract word limit and multiple revisions already implemented in response to previous reviewer comments, we were unable to further modify the Abstract without exceeding the permitted length. We believe that the implications of the very high heterogeneity are explicitly discussed later in the manuscript.

Action 1: No changes were made to the Abstract. If you still think that a qualification in the Abstract is necessary, we would be happy to revise it accordingly. Please advise on which part of the abstract to remove in order to not exceed the word limit.

Comment 2: “Introduction. The Introduction is well written and convincingly justifies the need for this systematic review. A brief early clarification distinguishing psychological symptoms from diagnosed psychiatric disorders would further strengthen the conceptual framing.”

Response 2: Thank you for your comment. We have added an explicit clarification in the Introduction part as well stating that this review focuses on psychological symptoms assessed by screening measures rather than on clinically diagnosed psychiatric disorders, and we have integrated this distinction into the stated aim of the meta-analysis.

Action 2: We have added the following: “This meta-analysis aims to consolidate available evidence and quantify the burden of psychological symptoms in patients with PC, as assessed by screening measures rather than clinically diagnosed psychiatric disorders.”

Comment 3: “Method: Search strategy: The search was conducted in MEDLINE, EMBASE, and Cochrane Library on May 22, 2022,119 and updated on March 3, 2025. There was no date or language restriction. How did you deal with the different language papers?”

Response 3: Thank you for pointing this out.

Action 3: We added the following sentence to address this point: […]for non-English articles, when none of the co-authors were fluent in the language, we used online translation tools to assess eligibility and extract relevant data.

Comment 4: “Eligibility Criteria: While the heterogeneity of psychological constructs was appropriately acknowledged, the Methods section does not clearly distinguish between clinically diagnosed psychiatric disorders and screening-based symptom measures. Explicit clarification that the outcomes represent psychological symptom burden rather than diagnostic prevalence would substantially improve methodological transparency.”

Response 4: We thank you for this comment and agree that an explicit clarification improves methodological transparency.

Action 4: We added the following sentence: “For the purposes of this review, outcomes were conceptualized as psychological symptom burden rather than prevalence of clinically diagnosed psychiatric disorders.”

Comment 5: “Statistical Analysis: Given the substantial methodological and conceptual heterogeneity across included studies, the authors should more explicitly justify the decision to pool prevalence estimates derived from different outcome definitions.”

Response 5: We thank the reviewer for this important comment. We fully acknowledge the substantial methodological and conceptual heterogeneity across the included studies (e.g.measurement tools). Despite these differences, we decided to pool prevalence estimates because all included outcomes were intended to capture the same underlying clinical construct. Although assessed using different instruments, the instruments used across studies were conceptually aligned and aimed to identify individuals meeting a comparable clinical or functional threshold rather than fundamentally distinct conditions.We explored the possibility of subgroup analyses; however, the available data were insufficiently detailed and too heterogeneous to form subgroups (except depression severity).

Action 5: We added the following sentence to the Discussion : "The pooled prevalence should be interpreted as an average estimate across heterogeneous operational definitions rather than a precise prevalence for any single measurement approach."

Comment 6: “Result: Study Characteristics: Fig 1. PRISMA Flowchart. Please provide reasons for excluding studies. A brief clarification in the main text explaining why certain studies were excluded from quantitative synthesis would aid readers.”

Response 6: Thank you for your suggestion, we agree with it.

Action 6: We made the following changes to the main text: “After assessing the full text, we excluded 194 articles, mainly due to ineligible study design (n = 12), no data on pancreatic cancer outcomes (n = 32), or no prevalence data (n = 44), resulting in 28 papers being used in the meta-analysis and 31 in the systematic review section.”

Comment 7: “Heterogeneity and Publication Bias: The implications of I² values exceeding 90% could be more clearly emphasised in the Results, not only in the Discussion.”

Response 7: Thank you for your comment. We agree that it needs to be more emphasised in the Result section too.

Action 7: We added the following sentence to our Result section: “High heterogeneity was observed across studies, with I² values exceeding 90% for all measured outcomes, indicating substantial variability between studies.”

We thank you again for your constructive comments. We look forward to your feedback and are happy to address any further questions or clarifications

Best regards,

Péter Hegyi and his team.

---

## [Decision Letter · Decision Letter 3]

16 Apr 2026

One out of four patients with pancreatic cancer experience psychological symptoms: a systematic review and meta-analysis

PONE-D-25-25313R3

Dear Dr. Péter Hegyi,

We’re pleased to inform you that your manuscript has been judged scientifically suitable for publication and will be formally accepted for publication once it meets all outstanding technical requirements.

Kind regards,

Cho Lee Wong, PhD

Academic Editor

PLOS One

Reviewers' comments:

Reviewer's Responses to Questions

**Comments to the Author**

1. If the authors have adequately addressed your comments raised in a previous round of review and you feel that this manuscript is now acceptable for publication, you may indicate that here to bypass the “Comments to the Author” section, enter your conflict of interest statement in the “Confidential to Editor” section, and submit your "Accept" recommendation.

Reviewer #9: All comments have been addressed

2. Is the manuscript technically sound, and do the data support the conclusions?

Reviewer #9: Yes

3. Has the statistical analysis been performed appropriately and rigorously? 

Reviewer #9: Yes

4. Have the authors made all data underlying the findings in their manuscript fully available?

Reviewer #9: Yes

5. Is the manuscript presented in an intelligible fashion and written in standard English?

Reviewer #9: Yes

6. Review Comments to the Author

Reviewer #9: Comments for Authors

Dear authors, you all did a great job. The following comments may improve the current manuscript.

1. Please provide details of excluded studies as supplementary file.

2. Please provide information about grey literature.

7. PLOS authors have the option to publish the peer review history of their article (what does this mean?). If published, this will include your full peer review and any attached files.

Reviewer #9: **Yes:** Israr Ahmad

---

## [Editor Report · Acceptance letter]

PONE-D-25-25313R3

PLOS One

Dear Dr. Hegyi,

I'm pleased to inform you that your manuscript has been deemed suitable for publication in PLOS One. Congratulations! Your manuscript is now being handed over to our production team.

Kind regards,

on behalf of

Dr. Cho Lee Wong

Academic Editor

PLOS One